# The refinement paradox and cumulative cultural evolution: Complex products of collective improvement favor conformist outcomes, blind copying, and hyper-credulity

Elena Miu [1,2,3,4‡]*, Luke Rendell[1‡]*, Sam Bowles[5], Rob Boyd[2], Daniel Cownden[6], Magnus Enquist[7], Kimmo Eriksson[7], Marcus W. Feldman[8], Timothy Lillicrap[9], Richard McElreath[3], Stuart Murray [1,10], James Ounsley[1,11], Kevin N. Lala[1]*

1 School of Biology, University of St Andrews, St Andrews, United Kingdom, 2 School of Human Evolution and Social Change, Arizona State University, Tempe, Arizona, United States of America, 3 MPI for Evolutionary Anthropology, Leipzig, Germany, 4 Department of Archaeology and Heritage Studies, Aarhus University, Aarhus, Denmark, 5 The Santa Fe Institute, Santa Fe, New Mexico, United States of America, 6 Ingrooves Music Group, Victoria, British Columbia, Canada, 7 University of Stockholm, Stockholm, Sweden, 8 Department of Biology, Stanford University, Stanford, California, United States of America, 9 Google DeepMind, London, United Kingdom, 10 School of Medicine, University of St Andrews, St Andrews, United Kingdom, 11 Marine Scotland Science, Freshwater Fisheries Laboratory, Pitlochry, United Kingdom

‡ These authors are joint first authors on this work.
* elena.miu@gmail.com (EM); ler4@st-andrews.ac.uk (LR); knl1@st-andrews.ac.uk (KNL)

## Abstract

Social learning is common in nature, yet cumulative culture (where knowledge and technology increase in complexity and diversity over time) appears restricted to humans. To understand why, we organized a computer tournament in which programmed entries specified when to learn new knowledge and when to refine (i.e. improve) existing knowledge. The tournament revealed a 'refinement paradox': refined behavior afforded higher payoffs as individuals converged on a small number of successful behavioral variants, but refining did not generally pay. Paradoxically, entries that refined only in certain conditions did best during behavioral improvement, while simple copying entries thrived when refinement levels were high. Cumulative cultural evolution may be rare in part because sophisticated strategies for improving knowledge and technology are initially advantageous, yet complex culture, once achieved, favors conformity, blind imitation and hyper-credulity.

## Author summary

Although social learning, culture, and traditions are found in animals, why humans seem to be the only species that builds on knowledge and technology over generations is still not fully understood. Here we organized a computer tournament in which programmed entries specified when to learn new knowledge and when to refine (i.e. improve upon) existing knowledge. We found a 'refinement paradox': while using refined behavior was beneficial to individuals, it was not beneficial to be the one doing the refining. Entries that

**Data Availability Statement:** All data and code is available online at https://osf.io/kw3v6/?view_only=5fa1998ffcc841fd895b40132309e55b.

**Funding:** Research supported in part by an European Research Council Advanced grant to KNL (EVOCULTURE, ref. 232823). The funders had no role in study design, data collection and analysis, decision to publish, or preparation of the manuscript.

**Competing interests:** The authors have declared that no competing interests exist.

refined selectively, only under limited conditions, did well while refinement levels increased, but once refinement was high simple entries that did not refine thrived. This result might explain why cumulative culture is rare in nature: sophisticated strategies for improving knowledge are initially advantageous, but once complex culture is common it pays to conform to the behavior of others and copy blindly.

## Introduction

Human culture is characterized by the accumulation and refinement of learned knowledge across generations, which is widely thought to underlie our species' success [1–7]. While social learning (*learning from others*) is common in nature, with well-understood benefits [1–8], complex cumulative culture (where knowledge and technology increase in complexity over time) is unambiguously found only in humans. Chimpanzees, whales, and songbirds have learned traditions [9–11], and some animals show iterative improvements in shared learned knowledge (e.g., 12), but only human culture has incontrovertibly generated advanced technologies that no individual could invent alone [2,3,5]. While high-fidelity transmission [3,13,14] and suitable demography [15,16] are probably important, the rarity of cumulative culture but ubiquity of social learning remains a mystery.

The evolution of social learning has been subject to intensive investigation [1,17]. Biological research, social science, and economic theory [18,19] all suggest that learning algorithms that allow individuals to identify the highest-payoff behavior should be advantageous. On the other hand, many researchers have been struck by human hyper-credulity [5,8,20–22] and the 'blind' (i.e., credulous, non-discerning) nature of human imitation (a.k.a. 'over-imitation' [23]). Curiously, people frequently adopt behaviors the advantages of which are difficult to understand, and often devise spurious explanations for these behaviors [5]. The advantages of improving pre-existing technology (here called 'refinement'), and how living in a technologically complex (i.e., 'refined') world affects learning strategies [24], have not been well-explored.

Analysis of these phenomena has been hindered by the methodological challenge of cumulative culture, an inherently complex and long-term process [7,25]. Laboratory investigations have been limited to simple tasks (e.g., making paper aeroplanes [26]), or specific datasets (e.g., computer code [25]), focusing on psychological mechanisms, and combining cultural elements [27], while theoretical treatments have focused on increments in the number of traits [28,29], improvement in a single trait [15], or abstract functional analyses [14]. While these studies offer some insights, new approaches for investigating the general and, in particular, large-scale properties of cumulative culture would be useful (but see [30]).

Tournaments have had success addressing such complex questions as they allow for the simultaneous assessment of a large number of alternative strategies, proposed by individuals from different backgrounds. Competition between simulated strategies composed by individuals from different academic backgrounds and theoretical expertise has generated novel findings relative to simulations conducted from a single theoretical framework. For instance, tournaments were effective for investigating the evolution of cooperation [31], and the advantages of copying [32]. In such open contests, anyone can submit strategies that will compete in a specified complex simulation environment.

## Tournament and simulation structure

To understand why cumulative culture is rare, we organized a competition (i.e. tournament) in which entrants submitted computational strategies (henceforth 'entries') that competed for

€25,000 prize money in a simulation environment that models *cumulative* culture. The tournament utilized the basic simulated environment of Rendell et al. [32,33] but was revised and extended as described below. We attracted a diverse range of entrants, mostly academics spanning departments like anthropology, biology, cognitive science, psychology, engineering, computer science, business, but also journalists, and a group of school pupils. Our entrants' countries of origin included Germany, USA, UK, Canada, South Africa, Switzerland, and the Czech Republic. Approximately 30 entrants were private individuals who did not offer background details. As in our previous study, the winners (DC, TL) were invited as co-authors after winning the competition.

## Entries

Each entry submitted by the above entrants consisted of an algorithm specifying how 'agents' in this simulated world would behave (see sample entry in S1 Supporting Information). Submitted 'entries' corresponded to a set of rules that specified when an individual would use a behavior it already knows to obtain a fitness payoff (EXPLOIT), when it would engage in trial-and-error learning (INNOVATE), when it would learn through observing other individuals (OBSERVE) and, in some simulations, when it should invest in improving a behavior it already knows (REFINE). Performing the right behavior was important, as fitness payoffs depended on how well exploited behaviors were matched to the current environment. However, learning was not free, as there was a time cost incurred whenever an individual learned or refined a behavior. Entries were tested against each other using evolutionary simulations.

## Environment

The environment was a 'restless multi-armed bandit' [18,32], with 100 'arms', each representing a different behavior (or technology) with its own single payoff. Each behavior had a payoff, initially sampled independently from an exponential distribution, such that many behaviors had small payoffs, and few behaviors had large payoffs. The term 'restless' indicates that the payoff of any given behavior could change over time. Multiarmed bandits are widely used to study decision-making in biology, economics, artificial intelligence, and computer science because they mimic a common problem faced by individuals who must make decisions about how to maximize payoffs. They represent problems in the real world, for instance, where there are many possible actions, only a few of which yield high payoffs; where it is possible to learn socially or asocially; and where the environment changes continuously [32].

Each simulated environment contained a population of 100 agents, who had to learn and use the 100 behaviors in order to accumulate payoff. The behavior of each agent was defined by one of the entries in the tournament. Each entry was a piece of python code or pseudo-code that took as input each simulation parameter and dictated when the agents would learn a new behavior or use a behavior they already knew in order to acquire payoffs. The simulation framework allowed for one of three extensions of a base model (see Materials and methods): either adding *cumulative culture* (where REFINE was an available move), *model-biased copying* (where non-random copying was possible) or *spatial structure* (where migration between demes was possible).

## Moves

Agents aimed to maximize payoffs by choosing between four possible moves (named EXPLOIT, INNOVATE, OBSERVE, REFINE) each round. The four moves were chosen to simulate learning (INNOVATE–individual learning, OBSERVE–social learning, REFINE–improving an already known behavior) and exploiting learned behavior (EXPLOIT). This was

the minimum set of moves that allowed us to assess the trade-offs between learning and exploiting in the context of cultural adaptation. All moves returned information about the payoffs associated with that behavior. Individual agents were born 'naïve' and needed to learn behaviors before they could use them.

INNOVATE represented asocial (e.g. trial-and-error) learning, through which an agent learned a new behavior at random from those it did not currently know. INNOVATE added that behavior and its exact payoff to the behavioral repertoire of the individual. Therefore, an agent's repertoire consisted of the behaviors they had learned, along with the associated payoffs for those behaviors at the time of learning. If an individual already had the 100 possible behaviors in its repertoire, it gained no new behavior from playing INNOVATE. In the *cumulative* case, the new behavior was acquired with refinement level 0. INNOVATE did not attempt to simulate the process of innovation, but rather provided a vehicle for investigating in which contexts such innovation might be advantageous by allowing computation of the cost-benefit consequences of playing that move. In this regard, INNOVATE represented any and all innovation processes. The fact that in our simulations a new behavior was chosen at random should not be interpreted as implying that innovation was reliant on random decision-making, but rather reflects our assumption that a new behavior was adopted that was distinct from behaviors pre-existing in the agent's repertoire. We are aware that academic fields vary in their use of the term 'innovate' and that potential confusion can arise because in some fields the use of this term is restricted to the first use of a behavior or technology in a population, while in others all instances where an individual devises a solution that is novel to it are described as innovations, even when others in the population have already devised that solution [34]. Here our use of INNOVATE matches the latter definition.

OBSERVE represented any and all forms of social learning. Again, our objective was not to simulate the learning process but rather to explore the circumstances under which social learning could prove adaptive. By playing it, individuals could observe, learn and estimate the payoff(s) of the behavior(s) being used by some number (the exact number being a simulation parameter $n_{observe}$, only relevant in the model bias extension) of other individuals in the same round. This knowledge was then added to the observing individual's repertoire. Only those individuals playing EXPLOIT in the same population as the observer were available to be copied. It was possible for an individual to OBSERVE a behavior already in its repertoire, in which case only the payoff recorded for that behavior was updated. The sequence in which agents were selected to move did not affect the availability of models for copying–all agents were required to submit a move, and OBSERVE moves were processed subsequently, so that all agents playing EXPLOIT were potentially available to OBSERVE.

OBSERVE was error prone with regard to both behavior and payoff. Each of the *nobserve* social learning events failed with a certain probability, and nothing was learned. The probability of failure was a parameter of the simulations, called *pcopyFail* (see Materials and methods for values used in simulations). For simplicity, we assumed that *pcopyFail* was unaffected by refinement level. Where social learning failed, the individual received no new behavior or knowledge of its payoff. Furthermore, the payoff estimate returned was a value drawn from a Poisson distribution with mean equal to the true payoff. This meant that larger values would be associated with larger errors.

REFINE enabled an individual to invest time in improving a behavioral act that it already knew (all agents were assumed capable of refining behavior, and all behaviors could be refined). Behaviors could represent, for example, use of tools, with refinement representing improvement of one tool type (see hammer technology example, Fig 1). Individuals playing REFINE had to specify which behavior from their repertoire they wished to improve. The result was an increase by 1 of the refinement level that individual knew for the selected

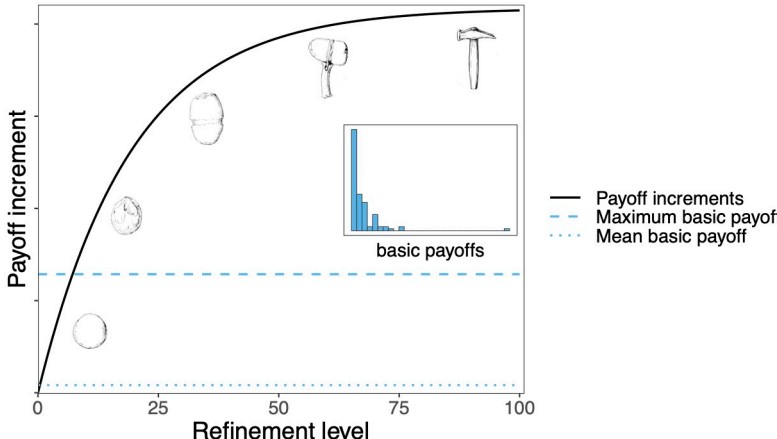

**Fig 1. Assumed relationship between refinement level and payoff increment, illustrated by refinements in hammer technology.** The payoff to a behavior is given by its basic (unrefined) payoff plus an increment that is a function of refinement level. Refining resulted in payoff increments (illustrated by the black curve) surpassing the highest basic payoff (dashed blue line) after approximately 10 refine moves. The inset illustrates an example distribution for the basic payoffs–most payoffs are low, and few payoffs are high.

behavior. Playing REFINE allowed agents to increment the refinement level ($r$) of a behavior they already knew by 1 'step' to achieve a higher payoff a maximum of $r_{max}$ times, assuming diminishing returns as $r \rightarrow r_{max}$. Our assumption that the relationship between improvement and payoff followed a diminishing returns curve is both empirically grounded [25] and is common in previous work [35,36].

The resulting payoff available to that individual for that behavior was equal to the basic payoff defined by the environment (which could change), plus an increment that was a function of the refinement level and was unaffected by basic payoff changes (see Materials and methods). Individuals did not know what refinement level they had achieved for that behavior, only that they had increased it by 1. They learned the new total payoff available for that behavior, without error. Refinement levels were zero when behaviors were learned through INNOVATE, but copied behaviors retained the refinement level of the behavior acquired through OBSERVE. Any behavior in the repertoire could be refined, irrespective of whether it was acquired through asocial (INNOVATE) or social learning (OBSERVE). Here refinement was treated as a purely asocial learning process, but in the real world, refinement could be a social activity [37]. When the environment changed, the base level payoff associated with a refined behavior changed, but the behavior did not change in refinement level. As for other learning moves, our analyses do not attempt to simulate the process of refining but rather to understand when investing in refinement would prove adaptive. REFINE therefore represents any and all refinement mechanisms.

Our decision to make the refinement level opaque to both the refiner and an observer reflects that refinement level is an abstraction, imposed for computational convenience, rather than a visible real-world property. In contrast, refiners and observers *were* able to detect the payoff to refined behavior when it was performed (albeit with some level of error in the latter case), and it is this information rather than refinement level per se that is most often salient in real world comparisons of alternative behaviors. Here, for simplicity, we assume that refining always leads to increments in payoff. While a reasonable approximation, we recognize that in the real world this assumption may not always hold.

EXPLOIT was the only move that resulted in a direct payoff to the acting individual. Note, EXPLOIT did not mean that another individual was taken advantage of, only that an individual had exploited its own knowledge to acquire a payoff. Individuals playing EXPLOIT had to specify which behavior they wished to deploy. An individual could only EXPLOIT behaviors it had previously learned. When an individual chose to EXPLOIT a behavior, the payoff it received was used to update the payoff recorded in its repertoire (that is, we assumed an individual could, by performing a behavior, update its knowledge of how profitable that behavior was).

INNOVATE, OBSERVE, and REFINE all carry the same opportunity cost—when agents made any of these moves they missed the opportunity to play EXPLOIT and gain the associated payoff. We did not impose any other costs or benefits to playing these moves, but recognize that in the real world additional costs (e.g., risks associated with innovation) and benefits (e.g., status increments associated with refinement) may operate. INNOVATE and REFINE allowed individuals to learn payoffs without error, while OBSERVE was error-prone—in practice there will be some level of error associated with asocial learning, but here the error associated with social learning can reasonably be construed as representing the difference in error associated with the two forms of learning.

## Environmental change

The environment was defined by the payoffs associated with each behavior, which changed stochastically. Environmental variation was simulated by changing behavioral payoffs, with probability $p_c$, per behavior per simulation round. The existence of environmental change meant that the payoff recorded for a given behavior related to when that behavior was learned, and if the payoff for that behavior had subsequently changed, then the payoff level that the individual had recorded in its repertoire could be wrong (in which case it would receive a different payoff from the one it had learned).

## Evolutionary dynamics

Evolution was implemented through a standard death-birth process in which individual agents died at random and were replaced by the offspring of individuals selected to reproduce with probability proportional to their fitness, which was measured by their accumulated payoffs. Agents inherited the learning strategy (i.e., tournament entry) of their parents (unless mutation occurred, in which case offspring were assigned an entry randomly selected from the others in the simulation, with probability 1/50). Each simulation run was initialized with a population of 100 naïve agents, all governed by a single entry. Through mutation, other entries could arise in this population. Mutation did not occur in the last quarter of each simulation.

## Score

Each entry in a simulation run was allotted a score based on the frequency of agents using that entry in the population in the last quarter of the simulation (see Materials and methods). Therefore, an entry's success was an indirect aggregate of its agents' payoffs. Note that while the agents only 'knew' their own repertoire, and their own history of behavior performed and fitness payoffs received, the designers of the entries (i.e., the entrants) had knowledge of the entire tournament procedure and could incorporate this information when devising their entries.

## Procedure

A simulation run proceeded as follows: each simulation started with 100 naïve agents from one entry and a set restless multi-armed bandit with 100 arms. Through mutation, a second entry (or several others, see Stages below) could enter the population. Agents used the four moves according to the instructions set by the entry that directed them in order to learn behaviors and obtain payoffs. Occasionally, agents died and were replaced proportional to their payoff by offspring who inherited their entry (unless mutation occurred). Occasionally, the environment changed. The frequency of agents belonging to each entry was measured in the last quarter of the simulation. The average of this frequency corresponded to that entry's score.

## Entry evaluation

We received 51 entries. The tournament was run in three stages. Stage 1 involved repeated contests between all pairs of entries (2550 total contests), in which a mutant entry invaded a resident population of 100 copies of the other entry. The simulation allowed for one of three extensions of a basic simulation framework (see Materials and methods): *cumulative culture*, *model-biased copying*, or *spatial structure*. All entries competed in all extensions. The top 10 performers in each extension, as measured by their frequency in the simulated population, progressed to Stage 2, where they competed simultaneously (melee-style) over a broader range of conditions within that single extension (three separate melees). The top 5 performers in each specialist extension progressed to Stage 3, where contests involved all three extensions at the same time. The stage structure was not intended to model a naturalistic process, but served as a heuristic device that allowed us to investigate each factor independently and combine them systematically. Here we focus on results from the cumulative culture extension, with other extensions shown for comparison.

## Results

Scores in the first stage varied from 0.06 to 0.86 (bounds: 0–1), indicating considerable diversity in effectiveness. The total amounts of learning (Regression estimate±s.e.: −0.798 ±0.078; learning is defined as the proportion of all INNOVATE+OBSERVE+REFINE moves, i.e. any move but EXPLOIT) and, in the cumulative extension, the choice of playing REFINE (−0.057 ± 0.035) were both key negative predictors of entry success. Consistent with analyses in non-cumulative settings [32], the proportion of learning that was social (0.379 ± 0.077, see Table A in S1 Supporting Information for definitions) was a positive predictor of higher scores (Table A in S1 Supporting Information). Successful entries minimized their time spent learning so as to maximize payoffs (i.e., time playing EXPLOIT; Fig 2A), with copying the most efficient way to learn even in the cumulative culture extension (Table A in S1 Supporting Information). In contrast, playing REFINE did not generally pay (Figs 2B and A in S1 Supporting Information).

Comparisons of learning rates among extensions showed that the opportunity to play REFINE (cumulative extension) was associated with an overall reduction in learning (Fig 2C and 2D), while in the extension that allowed for spatial structure learning rates typically increased relative to the other extensions. Learning rates were lowest in the cumulative extension, where typically a single behavior was highly refined; once this behavior was learned, it paid to EXPLOIT as it was virtually impossible for agents to acquire an alternative behavior with a higher payoff. Most (>80%) simulations achieved refinement levels with payoffs surpassing the highest basic payoff (i.e. unrefined payoffs; Fig B in S1 Supporting Information). Conversely, spatial structure increased learning rates because, on arrival in a new deme, entries tended to direct migrating agents, whose prior experience was outdated, to copy residents.

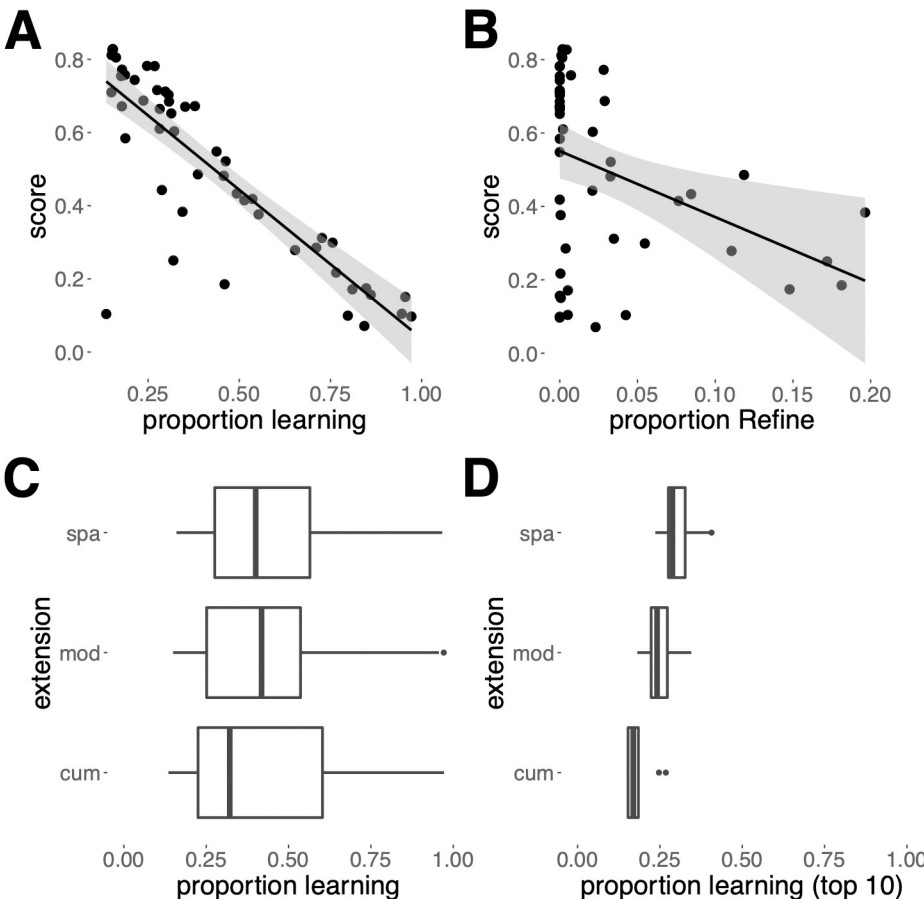

**Fig 2. Scores and learning in Stage 1.** Relationship between score and (**A**) the proportions of learning (INNOVATE +OBSERVE+REFINE) moves, (**B**) Score as a function of REFINE moves, averaged over each entry in Stage 1. (**C**) Distribution of the proportion of learning moves averaged by entry, for each extension over all entries and (**D**) for the top-ten best-performing entries.

These conclusions are supported by analyses of behavioral diversity (Figs 3 and F in S1 Supporting Information). In the absence of the opportunity to play REFINE (i.e. in model-based and spatial extensions), payoffs were more even (see Materials and methods for definition) and a new high-payoff behavior could be acquired through learning, allowing substantial diversity in 'behaviors' (i.e. behaviors that at least one individual in the simulated population used) and 'knowledge' (i.e. behaviors that at least one individual in the simulated population acquired and 'knew', but did not necessarily use with the EXPLOIT move) to be maintained within populations. However, in the cumulative extension, populations rapidly converged, through copying, on a very small number of persistently-performed behaviors, while other learned behaviors with lower payoffs were rarely performed or copied, leading to smaller population-wide repertoires, and the appearance of conformity. As refinement levels increased, diversity of both knowledge and behavior collapsed, while highly-refined behaviors (i.e. behavior with a high refinement level) persisted far longer than in other extensions. The most successful entries showed the same patterns in exaggerated form, implying this convergence on a small repertoire was adaptive (Figs G and H in S1 Supporting Information).

Our findings reveal a 'refinement paradox' in cumulative culture: it is better for everyone in the population to live in a world where payoffs are higher because of a history of refinement,

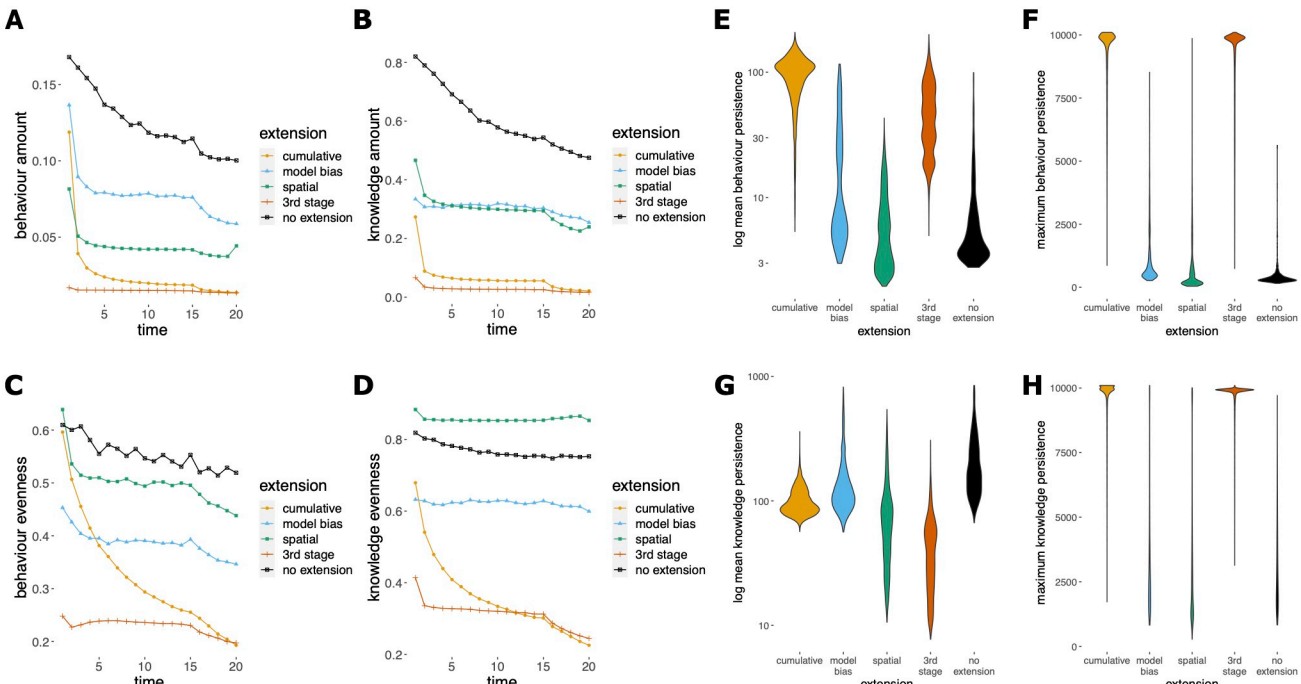

**Fig 3. Cultural diversity measured across extensions for Stages 2 and 3. (A-D)** Cumulative culture led to plummeting diversity in both the behaviors performed and known about, as populations converged on heavily refined behaviors, that **(E-H)** persist for long periods of time. 'Behavior' refers to the acts that the population was using at each timepoint, and 'Knowledge' refers to the acts present in the repertoire of at least one individual, but not necessarily used. 'Amount' captures the proportion of behaviors or knowledge known about within the population (i.e., mean proportion of possible behaviors used or known by at least one agent in each round in the last quarter of the simulations), 'evenness' measures the flatness of the frequency distribution (using Pielou's evenness index, see Materials and methods), and 'persistence' refers to the length of time the behavior or knowledge persisted in the population (i.e. mean or maximum number of rounds that a behavior within a population was exploited or that knowledge of it persisted within the population, without a break; see Materials and methods for more detail). Data for the 'No extension' case give a baseline comparison and are from 1,000 simulations using the top-ten tournament entries with randomly chosen parameter values.

but playing REFINE is not generally advantageous at the individual level [2] (Fig 4A and 4B and Tables B and C in S1 Supporting Information). Entries playing REFINE on average performed poorly, particularly in non-refined environments, where behaviors have not been improved (Fig 4A), and exhibited considerable variability. The paradox arises in the tournament because REFINE produces superior new knowledge that can be copied, and hence is a (non-rivalrous, non-excludable) public good, but the opportunity cost and diminishing returns of refinement mean it did not generally pay an individual to refine. Yet, in spite of the general disadvantage to playing REFINE, 'clever' entries could detect when refinement was worthwhile (i.e., gains would likely exceed opportunity cost) and could be advantageous (Fig 4B). In 60% of simulations in which refinement could occur, the maximum level of refinement was achieved (we call these environments that achieve the maximum refinement level 'refined' environments). The winning entry (called *farsightpolymorph*, submitted by DC and TL), together with entries placed 2nd-4th, all played REFINE, but under restricted conditions (Fig I in S1 Supporting Information). The cost of playing REFINE (and also OBSERVE, INNOVATE) is missing out on the payoff when playing EXPLOIT. By elevating payoffs beyond the base distribution, REFINE exacerbates that opportunity cost, and in refined environments, where behaviors have exceeded the maximum base payoff, the best entry is often 'blind copying' (a single OBSERVE followed by repeated EXPLOIT) as any observed behavior will likely have a high payoff. Conversely, in non-refined contexts, much of the behavior exploited has a

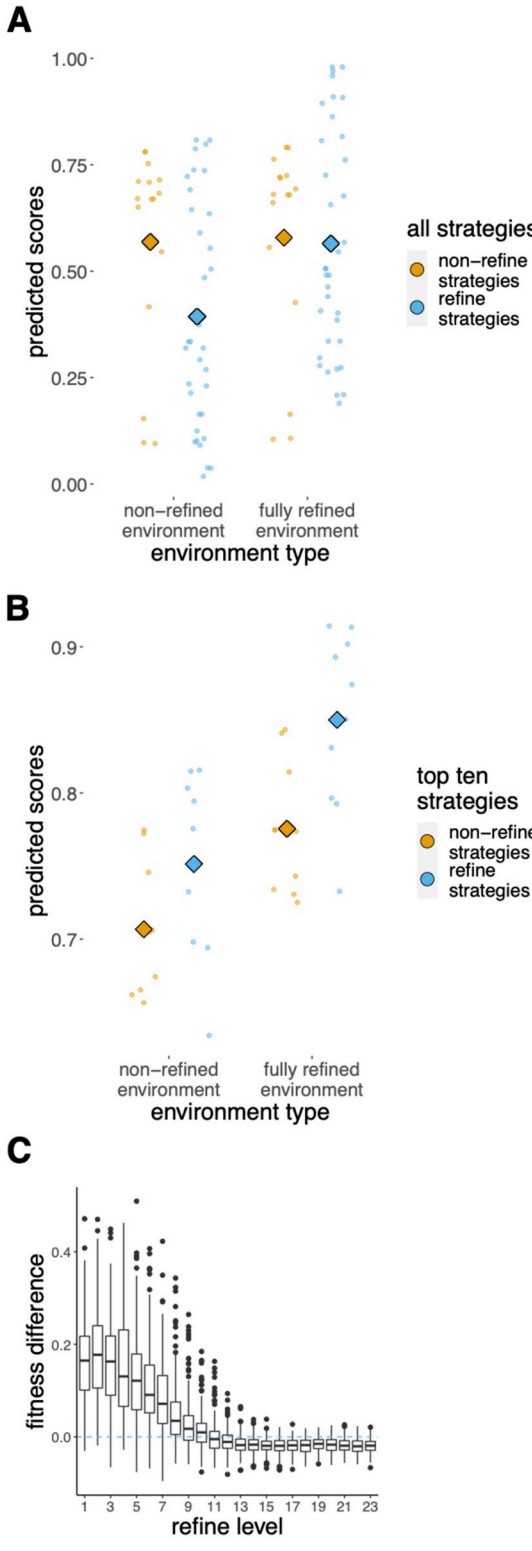

**Fig 4. The refinement paradox. (A)** and **(B)**. Predicted scores from a linear mixed model accounting for between-entry variation, using Stage 1 data, for (**A**) all entries, and (**B**) top-ten entries, that did and did not play REFINE, in refined compared to non-refined environments (Tables B and C in S1 Supporting Information). Top-ten entries used refinement strategically, achieving higher scores, and constructing maximally refined environments beneficial to all. (Circles indicate entry means and diamonds show group means, as predicted by the model. Environments were

defined as refined when a population reached the highest refinement level). (**C**) Relative fitness of 'clever' REFINE entries over a blind copier (OBSERVE-once-then-EXPLOIT-forever; see Materials and methods for details). 'Clever' entries had the advantage at low refinement levels, but were vulnerable to invasion at higher refinement levels.

low payoff, so learning or refining can find higher-payoff solutions. This may help to explain why humans are so reliant on social learning[8, 10] and often 'over-imitate'[24].

How can an entry know whether the environment is refined? Entries can remove uncertainty by creating a refined niche through a bout of playing REFINE. Refiners achieved a fitness advantage when the benefits of receiving the guaranteed higher payoff of refined behavior for longer than other agents (who eventually copy their behavior) out-weighed the opportunity costs of playing REFINE. Because of the diminishing returns of playing REFINE (rather than EXPLOIT), REFINE was more likely to occur in non-refined environments. Successful entries deployed REFINE in ways that maximized benefits and/or minimized costs (e.g., one entry, called *modes*, only refined when an agent was old, limiting non-exploitation costs). We compared the relative merits of 'clever' refiner (drawn from top-ten; Table D in S1 Supporting Information) entries to 'blind-copying' entries in environments with different levels of refinement. The simple heuristic of OBSERVE-once-then-EXPLOIT-forever, which does minimal learning, had a fitness disadvantage in populations at low refinement levels, but invaded a population of 'clever' refiners at high refinement levels (Fig 4C). One reason the winning entry, *farsightpolymorph*, was successful was that it essentially became this simple entry by ceasing to play REFINE when it estimated refinement levels were high in its current environment. In environments that were already 'refined' (see Table D in S1 Supporting Information for definitions) a much simpler learning entry could outcompete most entries that used refinement, which may help to explain why complex culture favors conformity. To the extent that the assumptions of the tournament hold, this implies, somewhat surprisingly, that blind copying may be more adaptive for humans than for other animals.

## Discussion

Claims have been made for cumulative culture in animals [12,38]; however, the evidence is limited, circumstantial and contested [39,40]. Our finding that refining entries generally struggled in unrefined contexts, but that strategically-deployed refining could be favored, helps to explain why cumulative culture is rare in nature. Plausibly, in addition to the evident challenges of refining, assessing the likely costs and benefits of refining to determine when refinement would be adaptive (as done by the best-performing entries) requires cognitive abilities that are rare in animal societies. Sophisticated communication (e.g., language, teaching) may also be important for effective copying of refined behavior [13,39,40].

Paradoxically, while humans clearly do have a capability for strategic refinement, technological advances leave human societies massively refined, in which case it is difficult for individuals to devise variants (i.e., to invent a new type of car or mobile phone) comparable in functionality to existing solutions. Results of the tournament suggest that humans may have constructed environments in which 'blind' copying was adaptive, providing a plausible evolutionary explanation for the 'hyper-credulity' [5,8,20], 'over-imitation' [23] and 'natural pedagogy' [41], documented by anthropologists and psychologists. This may also explain why refinement often arises through the workings of the 'collective brain' [37], and fits with recent evidence that collective decision making may have reduced demands on human cognition [42]. The extent to which changes in the archaeological record reflect individual-level innovation or collective decision making is debated [1,5–7,29,36], but both could have been possible.

Our finding that cumulative culture reduced cultural diversity is consistent with a recent empirical analysis of cumulative cultural evolution of online programming contests [25], as well as with some technologies thought to have exhibited 'boom-and-bust' patterns of diversity (personal computers, video games, cryptocurrencies), which a recent analysis attributes to a "dilution of expertise" [43]. Further work is necessary to establish whether a shortage of innovators or the absence of an incentive to innovate plays a greater role in real-world scenarios exhibiting a loss of cultural diversity. Such examples apart, the observed loss of diversity in the tournament may nonetheless appear surprising given the behavioral and technological diversity observed in most human societies. We suggest two possible explanations for this disparity. First, as the spatial extension of our tournament demonstrated, environmental heterogeneity and migration between demes promotes learning and maintains behavioral diversity. In the real world, spatial and other environmental variation in payoffs, combined with societal differences in values and utility functions, generate and preserve cultural diversity. Second, our analysis does not allow for recombination of behavioral variants, cultural exaptation (devising new functions for existing technology), a disproportionate valuing of novelty, or cooperation among innovators, all of which may be significant major sources of cultural diversity [5,8,14]. In contrast to the tournament setting, the continuous innovation that is characteristic of postindustrial societies may also lead to institutions that protect the interests of the refiner (e.g., intellectual property rights), but limit diffusion.

In designing the tournament, we had to make choices about its assumptions and its scope. Here we focused on material culture and technology as archetypal cases of cumulative cultural evolution that capture directly the adaptive value of culture. Whether other cultural domains like storytelling or music follow similar dynamics is currently under debate [44], but we imagine the results presented here could apply to other aspects of culture, with careful consideration. We have also made assumptions about the level of selection–here, payoff is individual-based, but the payoff structure could be modified to capture cooperative dynamics, by incorporating kin, or collaborative group effects. Our model provides a foundation on which such compelling questions can be explored in future work.

None of these additional factors undermine our general conclusions that strategic refinement generates complex technology, which in turn favors conformity and blind copying, observations strikingly evocative of human societies [5,6,20–23,37,40–42,45].

## Materials and methods

### Entry evaluation

**The competition was run in three stages.** Stage 1: *Single extension pairwise*. Valid entries ($n$ = 51) took part in each of the three sets of round-robin contests between all pairs of entries, with each set of contests involving one, and only one, of the three extensions (i.e., a model-bias set, a cumulative-learning set, and a spatial-structure set). A contest, say between entries A and B, involved exploring whether entry A could invade a population in which all individuals played only the entry of entry B, and vice-versa. Each contest involved replicate simulations, with each entry as the invader 50% of the time. In each simulation, a population of the dominant 'defender' entry was introduced, and run for 100 rounds in order to establish behavioral repertoires. Populations consisted of 100 individuals in total. Individuals died at random (with probability of 1/50 per round), and were replaced by the offspring of individuals selected to reproduce with probability proportional to their mean lifetime payoffs. Then, the second entry, 'invader', was introduced through mutation (i.e. an agent switched from using entry A to deploying entry B, with probability 1/50). Each such contest was replicated with 6 sets of parameters, twice with entry A invading B and twice with B invading A, for repeatability,

making 24 contests. The score of an entry in each simulation was the frequency of that entry in the population in the last quarter of the simulation (i.e. the proportion of agents in the population using entry A or B). The score of an entry in each extension was the average score of that entry across all the simulations in which it was involved. The parameter values chosen for these runs were probability of environmental change $pc$ = {0.001, 0.01, 0.1}, number of models available in the model-biased extension *nobserve* = {1,5}, maximum refinement level *rmax* = 100, probability of OBSERVE failing *pcopyFail* = 0.05 (Table E in S1 Supporting Information).

Stage 2: *Single extension melee*. From each *single extension pairwise* contest set, the ten highest scoring entries were entered into a *melee* with the same extension (i.e. simultaneous contest involving all ten entries). Within each of the three *melees* all entries competed simultaneously in multiple simulations across a broad range of conditions. The entries were given the opportunity to invade through mutation a standard defending entry, *innovateOnce*, which learned once asocially and then exploited that one behavior for the rest of its life. Entries were allowed to invade *innovateOnce* simultaneously. The score of an entry in each simulation was the average frequency of that entry in the population in the last quarter of the simulation. The winning entry for each extension was the entry with the highest average score within that extension contest. The second stage was run as two sets of simulations: one using parameter values drawn systematically from fixed sets of values, with 10 replicates each, and one drawing parameter values from preset exponential distributions, to explore the more broader regions of the parameter space. The systematic parameter values used were: $pc$ = {0.001, 0.005, 0.01, 0.05, 0.1, 0.2, 0.4} *pcopyFail* = {0, 0.01, 0.05, 0.1, 0.15, 0.25, 0.5} *nobserve* = {1, 2, 5, 10} *rmax* = {10, 25, 50, 100, 500, 1000}. The entry with the highest average score in each *single extension melee* was declared the winner of that single extension contest and won a prize (€5k). Thus by the end of stage 2 we had identified the (up to) three entries that operated most effectively in the specialized model-bias, cumulative and spatial contests.

Stage 3: *All extensions melee*. Finally, the five highest performing entries from each *single extension melee* competed in a further series of *melee* simulations in which all three extensions were run simultaneously. Score was calculated as above. Like Stage 2, this involved running simulations using a set of fixed parameters that was identical to the set used in stage 2, each repeated five times, as well as simulations using parameters drawn from exponential distributions. The entry with the highest average score in the *all extensions melee* simulations was declared the overall tournament champion, and won a prize (€10k).

## Simulation specifications

Each simulation contained either a single population, or, in the spatial case, three populations (demes) of 100 individuals, and ran for up to 10,000 rounds. A single round consisted of the following computational steps: (i) individuals were selected sequentially to choose a move until all individuals had played;(ii) individuals reproduced with probabilities proportional to their average lifetime payoffs; (iii) the environment changed, with a fixed probability $p_c$; and (iv) in the spatial case, some individuals migrated between populations.

## Environment

Each population had an associated environment, represented as a 'multi-arm bandit', wherein actors selected a behavior from a range of possible behavioral acts or technologies and received the payoff associated with that behavior. There were 100 possible behaviors, and the payoff for each behavior was chosen at the start of each simulation from an exponential distribution ($\lambda$ = 1; values were squared, then doubled, and finally rounded to give integers mostly falling in the

range 0–50 with occasional higher values). While we describe these activities as 'behaviors' or 'acts' they also represent the manufacture of tools and technology, each of which could be improved (i.e. playing REFINE). Each environment can be represented as a table with two rows associating each of the possible behavioral acts with the basic payoff received when that behavior is performed in that environment:

$$Behavior: \quad 1 \quad 2 \quad 3 \quad 4 \quad 5 \quad \ldots \quad 100$$

$$Payoff: \quad 4 \quad 0 \quad 17 \quad 1 \quad 7 \quad \ldots \quad 3$$

Payoffs were not constant, and the payoff associated with a given behavior changed between each round (i.e. generation) of the simulation with a probability, $pc$. New payoffs were chosen at random from the same probability distribution used to generate the original payoffs. The payoff for each behavior changed independently of the others, so that $pc$ also represents the average proportion of payoffs that changed in each round. In the *spatial* case, the three demes share the same set of 100 behavioral acts, but the initial payoffs for each behavior were drawn independently from the same distribution in all the demes. Subsequent changes in payoff were also independent in occurrence and magnitude across demes, although the rate of change, $pc$, was the same. Thus a behavior could pay, say, 4 in deme 1, 8 in deme 2, and 1 in deme 3. If that payoff changed in deme 1, this implied nothing about whether its value would change in other demes. In the cumulative case, payoffs also depended on the refinement level (see below).

Each individual agent had a behavioral repertoire, which typically contained a subset of the possible behaviors. Individuals were assumed to be born naïve; that is, they initially had an empty behavioral repertoire. Each individual's repertoire subsequently contained only those behaviors, and knowledge about their payoffs, that were acquired through asocial or social learning (i.e. INNOVATE or OBSERVE). The existence of environmental change meant that the payoff recorded for a given behavior related to when that behavior was learned, and if the payoff for that behavior had subsequently changed, then the payoff level that the individual had recorded in its repertoire could be wrong (in which case it would receive a different payoff from the one it had learned).

## Moves

In each simulation round, each agent could use a move. In all simulations, three options were available (INNOVATE, OBSERVE, EXPLOIT), while in simulations with the cumulative extension enabled, a fourth option (REFINE) was possible. We described how the four moves mainly operate in the main text.

In the case of OBSERVE, the number of models observed was a parameter of the simulation, termed *nobserve*. These models were selected at random from those available, except under the *model-bias* extension, when agents could select which to observe. If no individual played EXPLOIT in that round then there were no models and nothing was learned. Similarly, if the number of models was less than *nobserve*, only the available models were observed. It was possible for an individual to OBSERVE a behavior already in its repertoire, in which case only the payoff recorded for that behavior was updated. In the *cumulative* case, the behavior was observed at the same level of refinement as demonstrated by the model, meaning the observer could acquire that level of refinement and its associated payoff increment, for that behavior. The observer did not however know whether it had observed a refined behavior or not, nor the associated level of refinement.

In simulations with the *model-bias* extension, individuals electing to play OBSERVE were asked which of the available models they wished to copy, via the entry's *observe_who* function. In simulations without this extension, models were simply selected at random from the available pool, and this was also what happened if the entry did not define an *observe_who* function. Thus it was up to entrants to decide if they wanted their entry to make choices about whom to copy. The *observe_who* function was provided with the following information about every available model, giving various indexes of its performance: its age (in simulation rounds), its total payoff (the sum of all payoffs from EXPLOIT moves), the number of times it had been observed previously, and the number of offspring it had. We assumed this information, being social, was also error-prone, with error added in the same way as above—the returned value being drawn from a Poisson distribution with mean equal to the true value. This information was only made available once the decision to play OBSERVE had been made. The *observe_who* function returned the list of model information ranked in order of preference according to rules specified by the entry, and the first *nobserve* entries in this list became the learning models.

## Evolutionary dynamics

Evolution was simulated through a death-birth process. In the populations, individuals (i.e., agents) died at random with probability 0.02 per simulation round giving an expected lifespan of 50 rounds, and were replaced by the offspring of individuals selected to reproduce with probability proportional to their mean lifetime payoffs. The probability that individual $z$ reproduced was $Pz / \sum_i Pi$, where $Pz$ is the mean lifetime payoff of individual $z$ (that is, the sum of its payoffs from playing EXPLOIT divided by the number of rounds $z$ had been alive) and the denominator is the summed mean lifetime payoff of the population in that round. The mean lifetime payoff of an individual was unaffected by the number of offspring that it has produced. Entries were allotted a score based on their mean frequency in the last quarter of simulations.

While offspring were born with no behavioral acts in their repertoire they did, however, inherit the learning strategy (i.e. tournament entry) of their parents (unless mutation occurred). Mutation occurred with probability 1/50, and when it did offspring were allotted an entry randomly selected from the others in that simulation. Mutation was how other entries first arose in a population initially containing only a single entry. Mutation did not occur in the last quarter of each *melee* simulation.

In the *spatial* case, a number of individuals, *nmigrate*, were selected at random from each deme, and then each reassigned to a randomly selected different deme. Their repertoire was unaffected by migration–the agent still knew the same behaviors, at the same refinement levels in the *cumulative* case. They did not, however, know what the payoffs for those behaviors were in the new environment, because their knowledge was now outdated.

## Simulation parameters and cumulative payoff function

**Parameter *pc*.**   The probability that the basic payoff of a behavior changed in a single simulation round. In the round-robin stage of the tournament, simulations were run with a small number of *pc* values, drawn from the biologically plausible range [0.001–0.4]. In the *melee* stages, simulations utilized more values of *pc*, drawn from the same range (see Table E in S1 Supporting Information for exact values for each stage).

**Parameter *nobserve*.**   The number of models copied by an agent playing OBSERVE. In the pairwise tournament phase, *nobserve* took one of a small number of fixed values from the range [1–5]. Likewise, in the *melee* phases we ran simulations with $1 \leq$ *nobserve* $\leq 10$.

**Parameter *pcopyFail*.** The probability that social learning would fail on any given copying event. In the pairwise phase this was set to a single value, while in the *melee* phases simulations were run with values chosen uniformly in the range [0, 0.5].

**Parameter *nmigrate*.** The number of individuals chosen at random from each deme for migration (*spatial* simulations only). These were then reassigned to demes at random. In the pairwise stages, *nmigrate* was set to a single fixed value, while in the melee stages it was chosen from the range between 1 and 20.

**Parameter *rmax*.** The maximum allowed refinement level (*cumulative* simulations only). A cumulative payoff function defined how much would be added to the basic payoff of a behavioral act for a given refinement level *r*. This level, *r*, could range from 0 for a newly innovated behavior, up to *r*max. The payoff to a behavior was given by its basic (unrefined) payoff plus an increment *i*, where $i = \left( \frac{0.05}{1-0.95^{rmax}} \sum_{j=1}^{r} 0.95^{r-j} \right) pmax$. A graphical illustration of the relationship between refinement level and refinement increment is given in [Fig 1](). This represents a diminishing returns function but still offers payoff increments well in excess of the expected mean of the basic payoffs. Thus refining a behavior sufficient times resulted in payoffs that surpassed the highest basic payoff. In the pairwise stages, *rmax* was set to a single value, 100, while in the melee stages it took values in the range 10–1000. The limit *rmax* was introduced as a means to assay the circumstances under which refinement would approach a theoretical maximum. While diminishing returns is not the only conceivable function relating refinement level to payoff, in the absence of clear data we judge it to be plausible since, as technology advances, considerable investment (e.g. in training, specialist equipment and knowledge, industrialization) is required to achieve improvements.

## Procedure for entry

The tournament was widely advertised using posters, flyers, e-mail, listserves, conferences and social media (Facebook), from October 2011- February 2012, with a closing date of February 28 2012, With detailed information and entry requirements specified at www.lalandlab.st-andrews.ac.uk (link no longer active). Entries were computer code functions that took the specified data as arguments and returned a decision on which move to play (as well as such details as which individuals to copy, in particular cases). Entrants did not require knowledge of any programming language, as entries were required to be submitted either in code (Python v2.7) or 'pseudocode' (linguistic instructions breaking down how decisions are made into a series of mathematical and logical operations that could each be directly translated into a single line of computer code). The tournament was run in Python, and entrants familiar with that language submitted Python code directly, with an entry template provided on the tournament website. However, even if an entry were submitted as Python code, a pseudocode version had to be provided, to facilitate debugging. All submitted entries were also required to be accompanied by a brief prose description of how they were intended to function. Where entries contained coding errors or logical mistakes, for entries submitted sufficiently before the deadline, we attempted to contact the entrant(s) and invited them to revise their entry, provided they did so before the entry deadline.

Entries were required to specify up to two computational procedures: The first, termed *move*, took in information about an individual's life so far, and returned a decision about whether to INNOVATE, OBSERVE, EXPLOIT, or REFINE. The second, which only needed to be defined for entries that engaged with the model-based (*a.k.a. model-bias*) extension, was termed *observe_who*, and, in the event *move* decided on OBSERVE, took in information about the individuals that were available to copy and decided which of them to copy.

Individual agents were assumed to 'know' their own history of behavior performed and fitness payoffs received, allowing them to access and utilize this information. Each individual also had access to its own behavioral repertoire. We assumed that an individual could remember what it did over its lifetime, and how long it had been alive. Thus entries were provided with information on age, moves, behaviors exploited or learned, associated payoffs, and migration history.

Entries with functions that, on average, take more than 25 times as long as an example entry (provided to entrants) to reach a decision were disqualified. In practice, no entries were disqualified. Entries were also forbidden to access the disk or memory storage of the computer in any way beyond the information provided as input, so there was no way to store other information between rounds.

## Measuring cultural diversity

Cultural diversity was quantified as the number of behaviors present in the combined repertoires or expressed behaviors of all agents in the population (or deme) at a specific time, expressed as a proportion of the possible behaviors). We defined the 'knowledge' repertoire of a population as the combined repertoire of all the agents alive in the population at a particular point in time, while the population's 'behavior' repertoire was the set of behaviors being exploited across all individuals in the population at that time. For both, cultural diversity was quantified using metrics termed 'amount', 'evenness', and 'persistence' (after [33]).

*Amount* was calculated as the mean or median proportion of possible behaviors used or known by at least one agent in each round in the last quarter of the simulations, averaged over demes in the spatial extension.

*Evenness* represented the flatness of the frequency distribution of behavior patterns across the population, and was measured using Pielou's evenness index, $J = \frac{-\sum_{i=1}^{S} p_i ln p_i}{ln S}$, where $p_i$ traditionally represents the proportion of species $i$, and $S$ is the number of species [46]. In the tournament context, each 'species' $i$ represented a behavior, so $p_i$ was calculated as the proportion of agents using behavior $i$, and $S$ the number of behaviors. Thus maximum evenness was achieved when all possible behaviors were performed with equal frequency, while minimum evenness could represent the situation in which all agents performed the same single behavior.

*Persistence* referred to the mean, median, or maximum number of rounds that a behavior within a population was exploited or that knowledge of it persisted within the population, without a break. In the cumulative extension, refinement level was not differentiated–a behavior was treated as the same after it had been refined. We averaged each diversity measure over all simulations in a given extension, at every 1000 time steps, to see how the measure changed over time.

## REFINE as a measure of learning

For simplicity, we treated REFINE as a form of learning (alongside INNOVATE and OBSERVE). This can be justified on the grounds that a new variant of the behavior is achieved through refinement of an existing behavior, with a new payoff arising as a consequence. However, there is a qualitative difference between learning moves that bring a new behavior into the repertoire and learning moves that merely revise existing behavior and its payoffs, which needs to be recognized when interpreting findings related to the amount of learning across different extensions.

## Simulations to compare the relative merits of '*smart refiner*' and '*blind copying*' entries

We compared the relative merits of 'smart refiner' and 'blind copying' entries in environments with different levels of refinement. The 'blind copying' entry was OBSERVE-once-then-EXPLOIT ('observeExploit') while the "clever" entries used are 'Beancounter', 'combinator', 'epsilonGreedy', 'mayFlower', 'mlapd', and 'modes'. These 'smart' entries were chosen as six of the seven highest performing flexible entries. 'SenseAndAdapt' and 'ExploitEarlyNeverRefine' were not included as 'smart' entries, despite making the third round, as their performance was significantly worse than the others. We also excluded "farsightpolymorph" from this analysis as it is not easily categorized as a 'smart refiner' or 'blind copying' entry, being a high-performing refiner but otherwise behaving in a manner similar to 'observeExploit'. The analysis consisted of running a mini version of round three of the tournament but where the simulation stopped whenever the average refinement level of the performed behaviors on a given round was greater than a given stopping criteria. At that point the fitness of each agent in the simulation (namely, agent's mean lifetime payoff) was used to compute an average fitness score for clever and credulous agents, respectively. If innovateOnceExploit agents were left alive at the time of the stopping criteria, their average fitness was also measured and put in a class of its own (not plotted in figures). The data plotted in the figures are average 'smart refiner' or 'blind copying' entry fitness scores and the difference between them. Simulations were run until the average refinement level was greater than the designated refinement level and at that point the fitness of each agent in the simulation (i.e., agent mean lifetime payoff) was used to compute an average fitness score for 'smart refiner' or 'blind copying' agents, respectively.

## Supporting information

**S1 Supporting Information.** **Fig A.** Relationship between score and (a) amount of REFINE as a proportion of all learning moves; (b) proportion of just REFINE and INNOVATE learning moves; (c) proportion of INNOVATE moves; (d) proportion of OBSERVE moves and (e) proportion of learning moves that are OBSERVE (and not INNOVATE or REFINE). **Fig B.** Distribution of final maximum refinement level in the tournament (a) in all simulations, and (b) in simulations that include only entries that use REFINE. Refinement levels above the value of 8 typically lead to payoffs that exceed the maximum basic payoff (Fig 1). The maximum refinement level achieved across simulations varied substantially and showed a bimodal distribution. **Fig C.** Cultural diversity measures across extensions for Stage I. Cumulative culture leads to decreasing diversity in both the behaviors performed and behaviors known about, as populations converge on a small number of heavily refined, high-payoff behaviors. **Fig D.** Cultural diversity measures as a function of $p_c$ and $p_{copyFail}$ in Stage 2. Timelines of amount and evenness are presented as line charts, while the boxplots illustrate mean values for mean and maximum persistence. **Fig E.** Cultural diversity measures as a function of $r_{max}$ and $n_{Observe}$, in Stage 2. **Fig F.** Cultural diversity measures for one run of the cumulative extension in Stage II. (top) Proportion of agents who use and know each act, acts ordered by rank frequency, at increasing timesteps. (bottom) Behaviour and Knowledge evenness over time, with red dots indicating the same timesteps as above. The population very quickly converges on a small set of acts–in the last quarter of the simulation the population knows and uses only one act. **Fig G.** Amount and evenness of both behavior and knowledge in simulations that only include top ten entries (teal) and simulations with the rest of the 41 entries (red), in Stage I (cumulative extension). **Fig H.** Mean and maximum persistence of both behavior and knowledge, in simulations that only include the top ten entries (teal) and simulations with the rest of the 41 entries (red), in Stage I (cumulative extension). **Fig I.** Distribution of mean proportion REFINE moves per

simulation for each entry, ordered from the top left by score in Stage 1, in descending order. This plot only includes the 35 entries that used the REFINE move. Note the difference in scale on the x axis. The top four entries were also the top performers in Stage 3, which establishes that the best-performing entry used REFINE at low levels. **Table A.** Model averaged parameter estimates and summed Akaike weights (across an all-subsets model set of linear regression models with Gaussian error) for Stage 1 cumulative extension. Other extensions give similar findings (see Table B in S1 Supporting Information). The AIC_c best model (adjusted R-squared = 0.86) contained *pLearnRefine* and *pObserve*, as well as *meanPayDiffObserve*, *mean-BetweenLearn* and *doRefine*. The first two predictors were included in all of the 13 well-supported ($\Delta AIC_c < 2$) models; while the others were in 9, 6 and 7 respectively. **Table B.** Results from a linear mixed model predicting score as a function of the interaction between whether an environment was refined or not and whether an entry used refine or not, with a varying intercept for entry identity, using data for all 51 entries in the tournament. The model definition: score ~ environment type*entry type + 1|entry, assuming a non-refine entry in a non-refined environment as baseline. **Table C.** Results from a similar linear mixed model with a varying intercept for entry identity, using data for 20 entries, the top-scoring 10 entries that used REFINE and the top-scoring 10 entries that did not use REFINE. Model definition: score ~ environment type*entry type + 1|entry, assuming a non-refined entry in a non-refined environment as baseline. **Table D.** Definitions of new terms. **Table E.** Simulation parameters. (DOCX)

## Acknowledgments

We thank those who entered the tournament for their contributions, and Laurel Fogarty for helpful comments on an earlier draft.

## Author Contributions

**Conceptualization:** Luke Rendell, Kevin N. Lala.

**Funding acquisition:** Kevin N. Lala.

**Investigation:** Elena Miu, Luke Rendell, Daniel Cownden, Timothy Lillicrap, Stuart Murray, James Ounsley, Kevin N. Lala.

**Methodology:** Luke Rendell, Sam Bowles, Rob Boyd, Magnus Enquist, Kimmo Eriksson, Marcus W. Feldman, Richard McElreath, Kevin N. Lala.

**Project administration:** Luke Rendell, Kevin N. Lala.

**Supervision:** Luke Rendell, Kevin N. Lala.

**Visualization:** Elena Miu, Luke Rendell, James Ounsley.

**Writing – original draft:** Elena Miu, Luke Rendell, Kevin N. Lala.

**Writing – review & editing:** Elena Miu, Luke Rendell, Sam Bowles, Rob Boyd, Daniel Cownden, Magnus Enquist, Kimmo Eriksson, Marcus W. Feldman, Timothy Lillicrap, Richard McElreath, Stuart Murray, James Ounsley, Kevin N. Lala.

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
