## [Decision Letter · Decision Letter 0]

20 May 2024

Dear Miu,

Thank you very much for submitting your manuscript "The refinement paradox and cumulative cultural evolution: collective improvement in knowledge favors conformity, blind copying and hyper-credulity" for consideration at PLOS Computational Biology.

As with all papers reviewed by the journal, your manuscript was reviewed by members of the editorial board and by several independent reviewers. In light of the reviews (below this email), we would like to invite the resubmission of a significantly-revised version that takes into account the reviewers' comments.

We cannot make any decision about publication until we have seen the revised manuscript and your response to the reviewers' comments. Your revised manuscript is also likely to be sent to reviewers for further evaluation.

Sincerely,

Yamir Moreno

Academic Editor

PLOS Computational Biology

James O'Dwyer

Section Editor

PLOS Computational Biology

Reviewer's Responses to Questions

**Comments to the Authors:**

Reviewer #1: The central question in this article is of great interest to many of us researchers working on cultural and technological evolution: learning from others is natural, but the increase in knowledge and technological complexity over time is not. Why?

In the introduction, the article correctly identifies that the study of cultural complexity has been “hindered by the methodological challenge of cumulative culture, an inherently long-term process.” I concur that the artificial settings of controlled experiments (making paper airplanes, for example) have limited our understanding of cultural complexity. The lack of empirical studies is a significant barrier, as controlled experiments are insufficient to fully capture the complexity of empirical systems (see below). This, in my opinion, is a broader issue that affects the generality of lessons learned from theoretical studies of cultural evolution, including the present study. I'm concerned that the specific rules of the tournament may have biased the reported conclusions.

I must admit that I have struggled to interpret the findings of this study. This was most likely due to the tournament's somewhat abstract setting rather than a flaw in the paper methodology, which was well written and organized (although the analysis can be improved; see below). Aside from authoritative arguments, how can we be sure that the findings of these tournaments can be applied to real-world cultural evolution? For example, what is the rationale behind going through three different extensions? What actual system resembles a similarly staged selection process? An unambiguous method of demonstrating the validity of the chosen rules is urgently required. It would also be beneficial if the authors could motivate their research with a real-world scenario that is similar to the proposed tournament. This way, future expansions of the "tournament approach" will be built on a solid foundation.

The paper expands on a prior competition (“social learning strategies tournament”) that a subset of the authors organized in which 100 participants had to maximize their total payoff in a multiarmed bandit problem (Rendell et al. 2010). This approach was inspired by Robert Axelod's similar tournament in the 1980s, which investigated the evolution of cooperation. At each round, an individual must select one of three options/moves: "OBSERVE" to learn a behavior by observing another individual; "INNOVATE" to discover a new behavior on one's own; or "EXPLOIT" to obtain the current payoff associated with a known behavior. Individuals face a trade-off between receiving a payoff right away and learning a new and potentially better behavior that may result in payoffs later. The main conclusion of this tournament was that in an environment that is always changing, the more effective strategy is to periodically copy what others do instead of sampling the environment.

Moving on, determining the conditions under which cumulative cultural evolution can occur in this competitive environment is the primary contribution of the extended model. To achieve this, a fourth rule, 'REFINE', is added, which allows an individual to improve an existing behaviour. This new rule allows cumulative cultural evolution as other individuals can build on top of the improvements (“refinements”) made by other individuals. I found this rule to resemble more a kind of “parasitism” than a proper rule of complex cumulative culture. Many complex technologies require teamwork and cooperation among individuals, so one can imagine that the action of sharing information would be explicitly beneficial for both the demonstrator (the one executing “EXPLOIT” after doing some refinement actions) as well as the imitator (the one executing “OBSERVE”), as it happens in many open source software projects. How will the results of the tournament change if we make cooperation explicit?

Figure 1 is somewhat confusing to me because it implies some notion of progress, i.e., early forms of a technology yield higher pay-offs than more "finely tuned" recent versions. This contradicts the notion of an evolutionary process with no predetermined goal. On the other hand, I agree that such behavior may be valid for some (planned) technologies. Could you please provide the empirical data that informed the payoff curve for the Hammer case? Is there any empirical research showing that many technologies follow this curve? Or is it a heuristic assumption? To what extent do the results of the simulation depend on the specific shape of this curve?

I agree that "conformity, blind imitation, and hyper-credulity" are intrinsically associated with cumulative culture. The title of the paper somehow points to "complexity [collective improvement in knowledge] favours conformity," but it seems to me the causality runs in the opposite direction, that is, "conformity hinders complexity." Complexity is not a mechanism but a systemic property that emerges from an evolutionary process based on multiple mechanisms, including conformity biases. I would appreciate it if the authors could clarify this point.

A main result of the tournament is “finding that cumulative culture reduced cultural diversity” (line 283). Our most recent empirical study (which the authors did not appear to notice) supports this conclusion. In it, we measured how excessive imitation greatly decreased the cultural diversity of three large-scale real-world systems (see Duran-Nebreda, S., O’Brien, M. J., Bentley, R. A., and Valverde, S. (2022) “Dilution of expertise in the rise and fall of collective innovation,” Humanities & Social Sciences Communications, 9:365). Cumulative culture (along with complexity and diversity) can collapse due to the effects of “over-imitation” (line 48). We developed a theory that makes testable predictions about the loss of cultural diversity in real systems. For a robust (and system-independent) assessment of cultural trends, we have used three different measures of cultural diversity, including Kolmogorov complexity, which anticipates cultural collapses.

It would be beneficial to discuss (at least qualitatively) these two approaches, which appear to represent different hypotheses about the decline in cultural diversity. While the tournament attributes complexity to external drivers, such as "spatial and environmental variation in payoffs, combined with societal differences in values and utility functions, generate and preserve cultural diversity" (lines 287-290), our empirical analysis suggests an endogenous origin. This distinction appears to be the result of the tournament assuming that cultural evolution generates complexity primarily through competition rather than cooperation. The readers of the paper would greatly benefit from an explicit comment on this point in order to provide a fair representation of current approaches linking conformity with the loss of cultural diversity.

For example, what kind of functional curve corresponds to the evolutionary trajectories of cultural diversity (Figure 3)? They appear to be exponential, but we need to run a fitting test. It would be great if we could explain the fitting curve in terms of tournament parameters and measurements. In addition, some panels in Figure S4 show what appears to be a punctuated equilibria, with a sharp decay at around t = 15 in the curves of "mean knowledge amount" and "mean knowledge evenness." What is the origin of this shift? Many evolutionary algorithms display similar behaviors. In the context of cultural evolution, I am curious if this drop can be explained as a trade-off between conformity bias and the strength of selection (as suggested in Vidiella et al., 2022, "A cultural evolutionary theory that explains both gradual and punctuated change," J. R. Soc. Interface 19: 202220570).

Additional metrics could help to strengthen the arguments presented in the paper. The use of entropy-based measures of diversity has the disadvantage of producing similar values of evenness for very different distributions. To better understand how the system is behaving, we need to characterize the empirical probability distribution, p_i, or the proportion of agents using behavior i, which is used to compute the evenness (as described in lines 598-602). Please plot this distribution (at various points in time) in a figure so that we can better understand how the behavior/traits are distributed in the population. If the distribution has a long tail, such as p_i ~ i^{-\\alpha}, the exponent is a more accurate measure of diversity and can be linked to evolutionary mechanisms, such as conformity (Duran-Nebreda et al., 2022). In addition, the plots in panels F and H in Figure 3 suggest a heterogeneous distribution of the maximum and minimum knowledge persistence.

Finally, Figure S8 is very interesting, as it suggests a possible classification of the strategies based on the distance between the corresponding probability distributions of "REFINE moves." For example, "Grandma," "apc," "ShamelessExplorer," and "LifeLongLearner" have similar bimodal distributions and could be classified as a single group; however, "followTheMeans," "mayFlower," and "fiveYearPlan" have a similar long-tail distribution (power-law?) indicating a different class of behavior. The clustering of strategies could be summarized in a cladogram and discussed in the main text, allowing the reader to visually understand the breadth of strategies proposed.

Dr. Sergi Valverde,

Head of the Evolution of Networks Lab,

Institute of Evolutionary Biology (CSIC-UPF),

Barcelona

Reviewer #2: The manuscript investigates the phenomenon of cumulative culture. Through an exhaustive and well designed experiment conducted via a computer tournament, the authors delve into the refinement paradox, where refining behavior yields higher payoffs but does not generally pay off. Their findings shed light on the intricate dynamics underlying the evolution of knowledge and technology, revealing that while sophisticated strategies initially benefit individuals, complex culture ultimately fosters conformity and blind imitation. I really like this research and the conclusions. Nevertheless, I suggest the authors consider the optional subsequent comments:

1. Order of Concepts: The manuscript's structure is commendable, providing a comprehensive exploration of complex concepts. However, there are instances (for example, melee) where concepts are introduced before their definitions, which may momentarily confuse readers. Integrating brief definitions when introducing new concepts would further enhance the manuscript's clarity and accessibility.

2. Number of Moves: The experimental setup with four moves is intriguing, yet the rationale behind this choice is not explicitly discussed. Considering the significance of this decision, providing insight into why four moves were chosen and discussing the potential influence of this parameter would enrich the manuscript's depth and completeness.

3. Missing Reference: While the manuscript correctly references Render et al. 2010, there appears to be a missing reference with Render et al. 2011.

4. Preregistration: Preregistration of experiments enhances research transparency and reproducibility. I don't know if it was preregistered, although it does not affect my positive opinion about this research.

In summary, I commend the authors for conducting this groundbreaking research, which significantly advances our understanding of cumulative cultural evolution. The manuscript's meticulous methodology and insightful findings make it a valuable contribution to the field.

Reviewer #3: The paper addresses the emergence and rarity of cumulative culture. I believe

it's a great contribution to the field of cultural evolution. The authors

present a sophisticated and well-designed computer tournament to explore the

conditions under which cumulative culture -- characterized by increasing

complexity and diversity of knowledge and technology -- can evolve. This study

is not only innovative but also highly insightful, shedding light on the

refinement paradox and the intricate balance between knowledge improvement and

the benefits of simple imitation. The clarity of presentation is commendable,

although the complexity of the simulation experiment might be daunting for

readers not well-versed in such methodologies. Nevertheless, the findings are

profoundly exciting and add a critical layer to our understanding of cultural

evolution, particularly the nuanced dynamics that govern the rarity and

emergence of cumulative culture in nature. Below are some minor comments with

which I hope the authors can further improve their study.

* Comments on Clarity and Structure

- Make it clearer from the start how the different moves (INNOVATE, OBSERVE,

EXPLOIT) should be categorized. An important distinction is obtaining payoff

versus learning about payoff. Another is the accurate learning or error-prone

estimation of payoff. Explicitly defining these categories early on would help

the reader understand the roles and interactions of each move, improving the

overall flow and comprehension of the text.

- For the paragraphs describing the different moves, consider restructuring this

section to better prepare the reader for the various components of the model.

The current presentation is somewhat verbose and lengthy, making it

challenging to grasp how all the elements connect. The materials and methods

section is much clearer than the earlier explanations. Therefore, rearranging

the paper to present the more detailed descriptions before discussing the

results would provide readers with crucial details upfront and enhance their

understanding of the study.

- At the introduction of the INNOVATE move, it is unclear why this involves

accurate information about payoff. Similarly, it is unclear why the payoff is

estimated without error when observing. Clarifying these points would help in

understanding the rationale behind these assumptions and their implications

for the simulation results. This might however be resolved when rearranging

certain parts.

- It would be helpful to clarify from the start that this is a real-world

tournament. Due to the simulation context, I initially thought the designers

of the entries were also simulated. Adding a brief explanation about the

real-world nature of the tournament and the involvement of actual participants

would prevent this confusion. Additionally, providing more information about

the backgrounds of the designers of the entries could offer valuable context

and enhance the reader's appreciation of the tournament's scope and diversity.

* Comments on Methodology and Analysis

- While using Pielou's Evenness measure is an apt choice, it would be insightful

to compare it with alternative measures of evenness. This comparison would

provide a more complete understanding of the distribution of behaviors. See

Chao's work on Hill Numbers and Evenness for a more comprehensive presentation

(https://doi.org/10.1002/ecy.2852). Currently, to appreciate the point that

diversity decreases and, simultaneously, the remaining traits are distributed

less evenly, we need both Figure 3A and 3C. The evenness profiles proposed by

Chao would allow the authors to present this is a single graph.

- The authors show that social learning in general has a negative effect on the

diversity of behaviors, and that's also the case in the cumulative setting

with the refine move. It seems, however, that even without refine there is

still considerable decline in diversity as social learning increases. Would a

more comprehensive ablation study or model comparison help single out the

impact of different moves more precisely on observed diversity changes? I did

not find an explanation for this.

* Contextual Considerations

- In the simulation context, payoff benefits only the individual. In real-world

social contexts, payoffs can benefit more people (e.g., kin, friends). What if

the inner circle of an individual is enlarged to cover kin? This would make

the greater good on a smaller scale a more direct objective of an individual.

Exploring how social relationships and communal benefits influence payoff

dynamics could add an important dimension to the study's implications.

- The paper focuses on culture defined by material reflections such as tools and

technology. Can the authors also reflect on the emergence of other aspects of

culture that develop cumulatively but without clear payoffs, such as

storytelling, music, etc.? Including a discussion on non-material cultural

elements would broaden the scope of the study and highlight the diverse forms

of cumulative culture.

* Technical Comments

- The code is in Python 2.7, which is no longer supported. This makes it

challenging to run and test the code. Is Python 2.7 necessary, or could the

code be updated to more modern standards? Updating the code to a current

version of Python would enhance its accessibility and usability for other researchers.

- Provide an example in the section describing entries as computer code

functions. This would greatly aid in understanding how these functions

operate. An example would illustrate the practical implementation of the model

and clarify the decision-making process of the agents.

- Add information about cultural diversity quantification to caption of figure 3A

and B. This detail is crucial for understanding the results.

**Have the authors made all data and (if applicable) computational code underlying the findings in their manuscript fully available?**

Reviewer #1: Yes

Reviewer #2: Yes

Reviewer #3: Yes

PLOS authors have the option to publish the peer review history of their article (what does this mean?). If published, this will include your full peer review and any attached files.

Reviewer #1: **Yes: **Sergi Valverde

Reviewer #2: **Yes: **Carlos Gracia-Lázaro

Reviewer #3: No
---

## [Decision Letter · Decision Letter 1]

21 Aug 2024

Dear Miu,

We are pleased to inform you that your manuscript 'The refinement paradox and cumulative cultural evolution: complex products of collective improvement favor conformist outcomes, blind copying,and hyper-credulity' has been provisionally accepted for publication in PLOS Computational Biology.

Best regards,

Yamir Moreno

Academic Editor

PLOS Computational Biology

James O'Dwyer

Section Editor

PLOS Computational Biology

Reviewer's Responses to Questions

**Comments to the Authors:**

Reviewer #2: The revised version of this paper has improved significantly with all the raised issues addressed and, in my opinion, it can be accepted for publication.

**Have the authors made all data and (if applicable) computational code underlying the findings in their manuscript fully available?**

Reviewer #2: Yes

PLOS authors have the option to publish the peer review history of their article (what does this mean?). If published, this will include your full peer review and any attached files.

Reviewer #2: **Yes: **Carlos Gracia-Lázaro

---

## [Editor Report · Acceptance letter]

29 Aug 2024

PCOMPBIOL-D-24-00507R1 

The refinement paradox and cumulative cultural evolution: complex products of collective improvement favor conformist outcomes, blind copying, and hyper-credulity

Dear Dr Miu,

I am pleased to inform you that your manuscript has been formally accepted for publication in PLOS Computational Biology. Your manuscript is now with our production department and you will be notified of the publication date in due course.

With kind regards,

Anita Estes
